# Chromoblastomycosis: A case series from Eastern China

**Sujun Liu, Huilin Zhi, Hong Shen, Wenwen Lv, Bo Sang, Qiuping Li, Yan Zhong, Zehu Liu, Xiujiao Xia** [ID]*

Department of Dermatology, Hangzhou Third People's Hospital, Affiliated Hangzhou Dermatology Hospital, Zhejiang University School of Medicine, Hangzhou, China

* 804534095@qq.com

## Abstract

Chromoblastomycosis (CBM) is a chronic fungal infection of the cutaneous and subcutaneous tissues caused by brown pigmented fungi. *Fonsecaea monophora* is one of the most common pathogens of CBM in China. Most formal cases have been reported from Southern China, however, the infection is not uncommon in Eastern China where very few case series are available. To describe the clinical aspects of CBM, we report a series of 11 cases between 2018 and 2021 at a single medical center in Eastern China. The patients were predominately male (n = 9) and the disease duration ranged from 3 months to 20 years. Plaque type lesions were the most common clinical manifestations. There were 7 cases of mild forms and 3 cases of severe forms. Among the 3 severe cases, one case gave up treatment due to economic poverty; one case did not respond to a 1-year systemic treatmen; one case was cured by combination therapy of 10 months. Other cases were cured by treatment with antifungal agents. All cases of direct mycological examination were positive. All isolates were identified by morphology and sequencing of the the ITS regions of ribosomal DNA, Ten were *F. monophora* and 1 was *Cladophialophora carrionii*. All cases had been evaluated at other clinics, where 8 cases were misdiagnosed as other diseases. As a neglected tropical disease (NTD), CBM is still a major challenge in the field of dermatology, especially in its severe clinical forms. As an effective and simple diagnostic method of CBM, direct microscopic examination should be further promoted in rural hospitals.

## Author summary

Chromoblastomycosis is a neglected fungal disease that mainly affects low-income agricultural workers and rural population. The high misdiagnosis rate of chromoblastomycosis is a major challenge in dermatology, especially for rural hospitals. With the increase in the disease severity, infections may bring significant limitations to labor activities. Severe lesions tend to respond slowly or even become non-responding to antifungal drugs. A direct microscopic examination can find pathogens with typical characteristics; so, this technology must be promoted and mastered through short-term training in rural hospitals. The disease is an implantation mycosis, and therefore, wearing protective clothing

**Data Availability Statement:** All relevant data are within the manuscript.

**Funding:** This work was supported by a grant from the Hangzhou Science and Technology Bureau (http://kj.hangzhou.gov.cn) grant 202004A17 to

ZHL. The funder had no role in study design, data collection and analysis, decision to publish, or preparation of the manuscript.

**Competing interests:** The authors have declared that no competing interests exist.

and shoes are effective measures to prevent these diseases. The treatment choice and results depend on the patients' lesion grade. Mild cases respond well to systemic agents, whereas high doses and a long treatment and combination therapy are the first treatment choices for severe forms. This study presents clinical features, mycological findings, pathogens, treatments, and outcomes of CBM from Zhejiang, Eastern China.

## Introduction

Chromoblastomycosis (CBM) is an implantation mycosis, consisting of chronic cutaneous and subcutaneous lesions that develop at the site of previous transcutaneous trauma [1]. CBM is strongly associated with agricultural activities and the lack of protection, which further underscores the occupational nature of this disease [2]. It can lead to chronic persistent infections and may cause an incapacity for labor in some severe clinical forms. Although known for 100 years, CBM still poses a therapeutic challenge to clinicians due to its recalcitrant nature and common relapse after treatment [3].The species of the genus *Fonsecaea* are the most common etiological agent of CBM, and comprises four associated species: *F. pedrosoi*, *F. monophora*, *F. nubica* and *F. pugnacious* [4–7].*Cladophialophora carrionii* was the most common causative agent in the north of the Mainland China, and *F. monophora* and *F. pedrosoi* were the two most common agents in the southern part of Mainland China [8]. A comprehensive evaluation using molecular sequencing data showed that *F. monphora* is the most prevalent pathogen of CBM in Guangdong, Southern China [9]. We have carried out a clinical research project on cutaneous and subcutaneous infectious diseases at Department of Dermatology of Hangzhou Third People's Hospital between 2018 and 2021 and found 11 cases of CBM. To evaluate the clinical characteristics of CBM in Zhejiang, Eastern China, we analyzed their clinical features, mycological findings, pathogens, comorbidities, treatments, and outcomes.

## Methods

### Ethics statement

This study was approved by the ethics committee of Hangzhou Third People's Hospital and the study participants were informed about the study procedures and written informed consent was obtained.

A prospective descriptive study of patients with clinically suspected CBM or other important cutaneous and subcutaneous infectious diseases was performed at the medical mycology laboratory in the Hangzhou Third People's Hospital between January 2018 and December 2021. Age, sex, and occupation, and clinical data, such as comorbidities, traumatic history, duration of the lesions, and clinical type of skin lesions were recorded. The diagnosis of these cases was made by clinical observation, histopathology, mycological examination and nontuberculous mycobacteria (NTM) culture. The diagnosis of CBM was based on the definitions described and proposed by Carrión in 1950; in which the lesions are divided into five different types (verrucous, nodular, plaque, cicatricial and tumoral) [10]. CBM lesions were graded according to the criteria proposed by Queiroz-Telles et al [11]. This study was approved by the ethics committee of our hospital and the study participants were informed about the study procedures and written informed consent was obtained.

## Sample colleciont

We referred to the flowchart for laboratory diagnosis of sporotorichosis that was proposed by Orofino Costa et al [12], and developed a standardized operation process. For open lesions, including ulcer or abscess with fluctuation, tissue fluid was taken for microscopic examination and culture. For closed lesions without fluctuation, only tissue culture was performed. A sterilized lancet was used for scraping the surface of the lesions, principally those that were highly pigmented and known as black dots. Pus samples were collected by squeezing the skin lesions. In most cases, skin biopsies were performed and portions of the specimens were fixed in 10% neutral buffered formalin, embedded in paraffin, sectioned, and stained with hematoxylin & eosin (H&E) and periodic acid-Schiff (PAS) or acid-fast staining. Other fresh portions were directly inoculated onto Sabouraud dextrose agar (SDA) and Lowenstein-Jensen solid medium.

## Etiological detection

KOH or fluorescent staining preparations of scales and pus that were obtained from the lesions were examined under a microscope for muriform bodies or septate brown pigmented hyphae, hyaline spores or hyphae. Separate scales or pus samples were used for fungal or NTM culture. Portions of skin biopsies and debris were inoculated onto culture media to recover the etiologic agent. The primary isolation of the fungus was performed on agar slants of SDA containing chloramphenicol (CMP, 0.125 g/l) and that were incubated at 25˚C for two weeks. Another primary isolation of the NTM was performed on Lowenstein-Jensen solid medium and incubated at 25˚C for four weeks. The fungal isolates were initially identified by slide-culture microscopy and NTM isolates were primarily identified by matrix assisted laser desorption ionization time of flight mass spectrometry (MALDI-TOF MS) (Bruker Daltonik MALDI Biotyper). The final identification of the isolated agent was performed by sequencing of the ITS 1 and ITS 4 regions of rDNA or 16S rRNA and compared with sequences deposited in GenBank by Blast program.

## Results

In this study, a total of 100 cases of important cutaneous and subcutaneous infectious diseases were found, including 45 cases of sporotrichosis, 29 cases of NTM infection, 11 cases of CBM, 7 cases of Majocchi' s granuloma, 4 cases of hyalohyphomycosis, 2 cases of candida infection, 1 case of phaeohyphomycosis and 1 case of *Talaromyces (Penicillium) marneffei* infection (Fig 1). A total of 11 patients with CBM came from Zhejiang province and their clinical data are summarized in Table 1. The age ranged from 39 to 83 years (mean of 62.9 years). There were six farmers, three carpenters, one gardener, and one retired teacher who planted a variety of flowers at home. The duration of CBM ranged from 3 months to 20 years (mean of 6.5 years), but most of the patients suffered from the infection for more than 1 year (75%). Six patients (54.5%) either could not recall or gave a negative history of trauma prior to their manifestations. Among five cases with a history of trauma, three cases were caused by a wooden spike prick and two cases caused by insect bite. Diabetes was the most common comorbidity. Three patients (27.2%) had diabetes, two patients had chronic kidney disease, one patient had a diffuse pulmonary interstitial fibrosis with hypertension, and one patient had heart disease. In 63.6% of cases (7/11), the lesions were localized to 1 site and the most common afflicted site was the upper limb (7 cases), followed by the lower limb (2 cases), the buttocks (1 case), and the trunk (1 case). The most common clinical variety of described lesions was plaque (63.6%, 7/11) followed by verrucous (18.2%, 2/11), nodular and cicatricial (9.1% each). In addition, according to their severity, lesions' grade showed

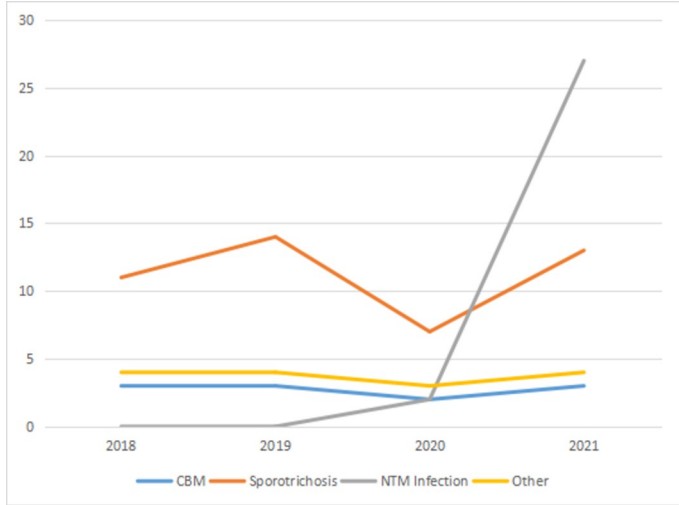

**Fig 1. Number of cases with important cutaneous and subcutaneous infectious diseases between 2018 to 2021.**
Note: We carried out NTM culture at the end of 2020.

three cases of severe form (Fig 2A–2C), especially case 1 who had lost the capacity for labor, seven cases of mild form (Fig 3A, 3C and 3E), and one case of moderate form. All patients reported that pain worsened as the course of disease increased. Before their referral to our clinic, all cases had been evaluated at other clinics, eight cases were misdiagnosed as other types of diseases, of which sporotrichosis and eczema were the most common diagnoses, followed by lupus vulgaris and neurodermatitis.

All cases of direct mycological examination were positive, and skin scrapings or pus that were treated with 10% KOH revealed muriform cells with or without germinated hyphae (Fig 4A and 4B). Histopathology proved the characteristic brownish thick-walled sclerotic bodies being demonstrable in 5 cases either within or outside the giant cells (Fig 4C). Pathogens were isolated from 11 cases. One pathogen were identified as *C. carrionii* (Fig 4D) and ten as *F. pedrosoi* (Fig 4E and 4F) based on their morphological characters. Based on the sequences of the ITS1-5.8S-ITS2 region of rDNA, 10 *F. pedrosoi* isolates reidentified as *F. monophora*.

**Table 1. Clinical and fungal results of eleven cases of chromoblastomycosis.**

| Case | Date | Age(years)/sex | Occupation | trauma | Clinical features/site/grade | Duration(m) | Treatment regimen | outcome | Pathogen |
|------|------|----------------|------------|--------|------------------------------|-------------|-------------------|---------|----------|
| 1 | 2018.2 | 61/M | Farmer | No | Verrucous/right ankle/severe | 180 | Lost to follow-up | No | *F. monophora* |
| 2 | 2018.3 | 68/M | Carpenter | Yes | Cicatricial/left forearm/moderate | 84 | TBF (0.25 g/day) for 5 months | Cured | *C.carrionii* |
| 3 | 2018.8 | 39/M | Farmer | No | Verrucous/buttocks/severe | 240 | ITZ (0.2 g/day) for 12 months | Failure | *F. monophora* |
| 4 | 2019.1 | 56/M | gardener | Yes | Plaque/left forearm/mild | 12 | ITZ (0.2 g/day) fo r4 months | Cured | *F. monophora* |
| 5 | 2019.7 | 62/M | Carpenter | Yes | Plaque/trunk/severe | 168 | ITZ (0.2–0.4 g/day) for 10 months + PDT | Cured | *F. monophora* |
| 6 | 2019.10 | 71/F | Farmer | No | Plaque/right leg/mild | 18 | ITZ (0.2 g/day) for 3 months | Cured | *F. monophora* |
| 7 | 2020.3 | 83/M | gardener | No | Plaque/left wrist/mild | 5 | ITZ (0.2 g/day) for 3 months | Cured | *F. monophora* |
| 8 | 2020.5 | 70/F | Farmer | No | Plaque/right forearm/mild | 120 | ITZ (0.4 g/day) for 3 months | Cured | *F. monophora* |
| 9 | 2021.3 | 57/M | Farmer | No | Plaque/left wrist/mild | 18 | ITZ (0.4 g/day) for 5 months | Cured | *F. monophora* |
| 10 | 2021.5 | 68/M | Farmer | Yes | Nodular/left forearm/mild | 3 | ITZ (0.2 g/day) for 3 months | Cured | *F. monophora* |
| 11 | 2021.8 | 57/M | Carpenter | Yes | Plaque/right wrist/mild | 8 | ITZ (0.4 g/day) for 4 months | Cured | *F. monophora* |

F, female; M, male; ITZ, itraconazole; TBF, terbinafine; m, month; PDT, photodynamic therapy

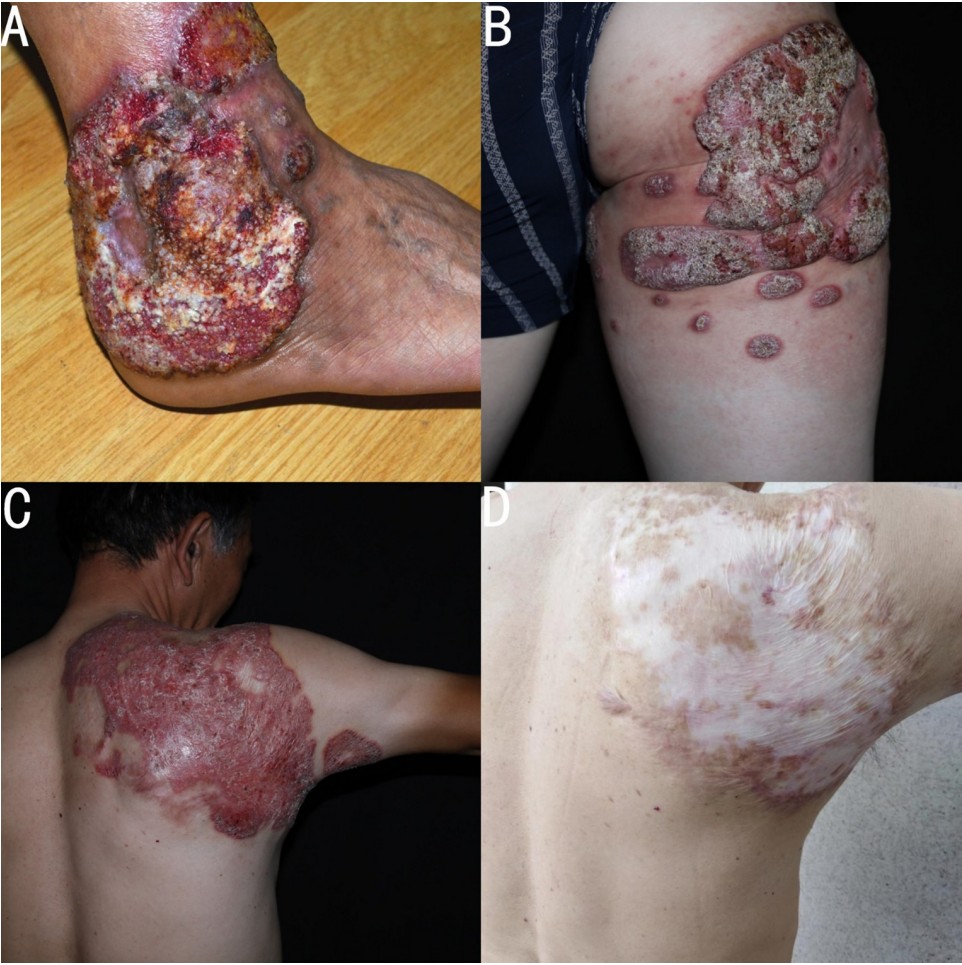

**Fig 2.** Severe form: verrucous skin lesions on the right ankle in case 1 with a loss of labor activities (A). Verrucous lesions on the buttocks in case 3. The lesions were unresponsive to itraconazole (200 mg per day) for one year (B). Clinical picture of case 5 with extensive erythematous plaque lesions (C), clinical cure after 10 months of combination therapy (D).

Finally, ten cases received systemic treatment, and case 1 gave up treatment due to economic poverty. The seven mild cases were cured by treatment with itraconazole 200–400 mg per day, with a treatment course that ranged from 3 months to 5 months (Fig 3B, 3D and 3F). Case 2 was cured by treatment with terbinafine 250 mg per day for 5 months. However, it should be noted that treatments of case 7, case 8 and case 9 have beed briefly interrupted due to the interference of comorbidity. Case 3 had been treated with itraconazole (200 mg per day) for one year, but no improvement was observed. Currently, case 3 is receiving a new treatment plan. Case 5 was cured by combined itraconazole and photodynamic therapy for 10 months (Fig 2D). In our study, the total rate of complete remission was 90% (9/10) in our study.

## Discussion

In China, chromoblastomycosis is primarily caused by *C. carrionii* in the north and by *F. monophora* and *F. pedrosoi* in the southern and eastern parts of the country [13], especially in Guangdong Province and Shandong Province of Mainland China [8]. *F. pedrosoi* and *C. carrionii* infections are normally observed in tropical and subtropical areas of endemicity around

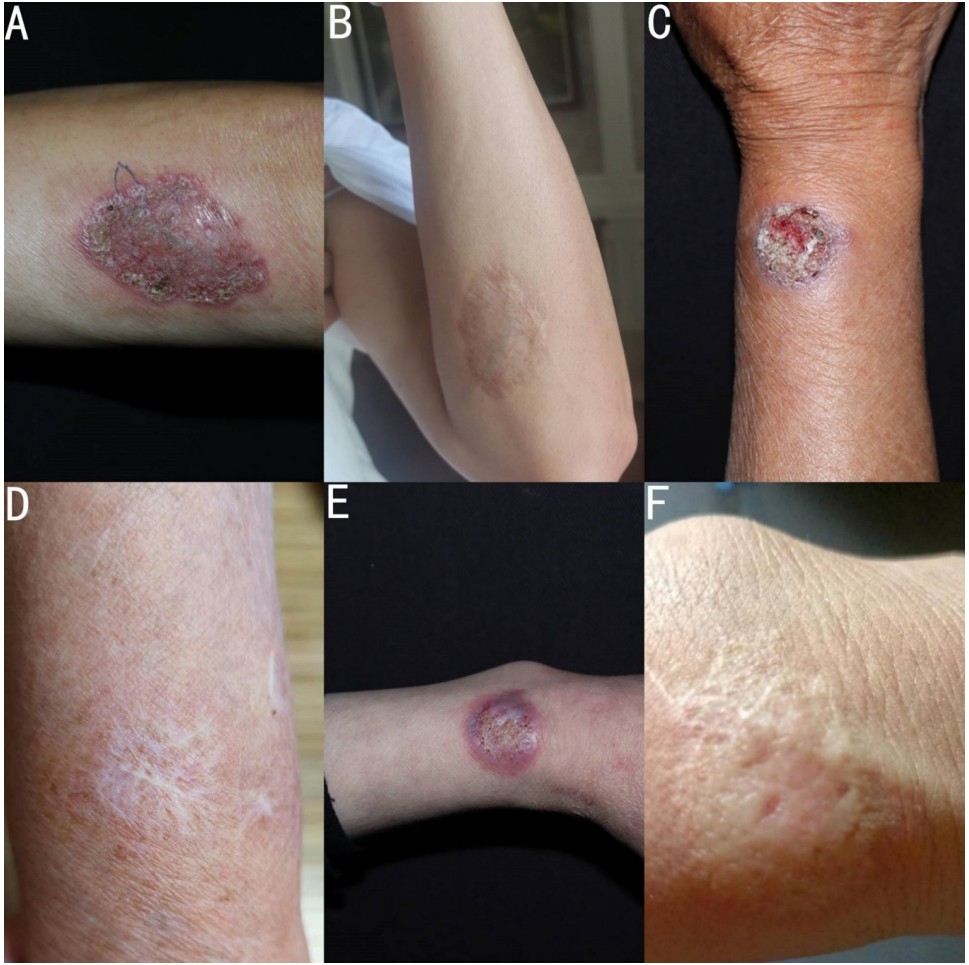

**Fig 3.** Mild form: Clinical picture of case 4, before treatment (A), and during 30 months of post-treatment follow-up (B). Clinical picture of case 8, before treatment (C), and during 14 months of post-treatment follow-up (D). Clinical picture of case 9, before treatment (E), and during 8 months of post-treatment follow-up (F).

the world [14]. These are brown pigmented fungi living as saprophytes on plants or vegetable debris in the soil [15]. Zhejiang is located in the southeast coast of China and in the middle of a subtropical zone, with monsoon humid climate. *C. carrionii* occurs in semiarid areas, whereas *F. pedrosoi* is associated with humid climates [14]. In terms of geography and climate, Zhejiang is suitable for the survival of these brown pigmented fungi, especially the genus *Fonsecaea*. Molecular phylogeny has facilitated the identification of cryptic species within the *F. pedrosoi* species complex. The isolates identified as *F. pedrosoi* morphologically now belong to *F. pedrosoi sensu strico*, *F. monophora* [16], and *F. nubica* [7]. *F. pedrosoi* and *F. nubica* are strictly associated with chromoblastomycosis, whereas *F. monophora* is also involved in phaeo-hyphomycosis of the brain and other organs [7]. In clinical practice, both *F. pedrosoi* and *F. monophora* are easily misidentified because both have similar morphological features. Although it may have limitations, the ITS region can confirm the species in most of the circumstances. These limitations may be overcome by the sequencing of other loci and/or by using other molecular methods. The combination of *CDC42*, *ACT1*, *BT2*, lactase (*Lac*), homogentisate (HmgA), and polyketide synthase (*PKS1*) (the first three genes are the most common) with ITS sequencing can increase the accuracy [17]. Based on our study, *F. monophora* is

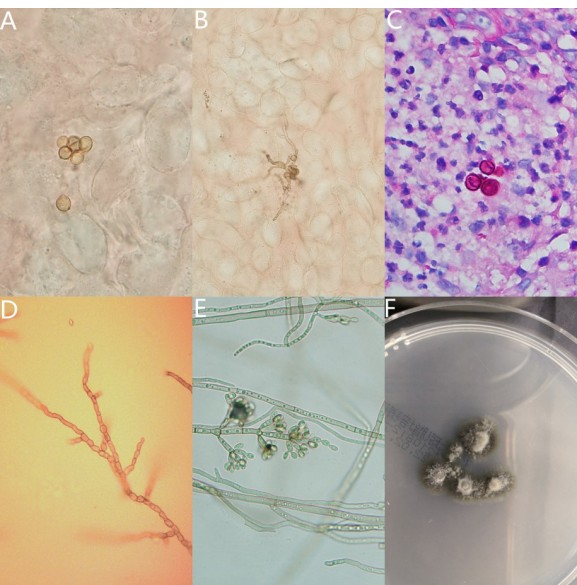

**Fig 4. Mycological features of CBM.** Pigmented muriform cell with obvious septation (A) and brown muriform cell with germinated hyphae (B) (10% KOH, ×400). Mixed granuloma with sclerotic body (PAS, ×400) (C). Slide culture on potato dextrose agar at 25˚C on day 8 of *C.carrionii* (D) and *F. monophora*(E) (×400). *F. monophora* colonies on SDA at 25˚C on day 7 (F).

the dominant pathogen in CBM cases found in Zhejiang. *F. monophora* isolates exhibited high genetic diversity [18]. and we hypothesized that many *Fonsecaea* isolates might have been mis-identified due to the lack of molecular tools in the past. Thus, a long-term surveillance and molecular identification of all species in *Fonsecaea* isolates from CBM cases are necessary for epidemiology.

In agreement with previous literature, we found that the identified CBM cases are prevalent among old male farmers. Chromoblastomycosis is considered an occupational disease world-wide that affects farm laborers, gardeners, lumberjacks, vendors of farm products, and other workers exposed to contaminated soil and plant materials [14]. On rare occasions, wood fragments containing muriform cell-like structures were histopathologically observed in patients with CBM [19,20]. In our series, carpenter was the second most common occupation. CBM is usually gradual and indolent in nature. As with sporotrichosis, CBM is difficult to eradicate, GRAPH 1 suggests that CBM has been in a stable sporadic state. At the time of presentation, the duration of the disease varied from 20 days to 35 years [3]. Most of the studied cases were chronic, with a mean duration of 6.5 years. The lower limbs are the areas of the body that would be most likely to be in contact with fungi contaminated material [21]. In most series, there has been a predilection for the lower legs [2,3,21,22]. However, in Australia, the upper limbs were the most affected parts by CBM [23], which is also observed in our cohort, where the hand, wrist, and forearm were predominantly affected. The difference might be attributable to differences in economic structures among these countries. Rural workers in Eastern China tend to wear protective clothing and shoes that cover the legs but not the hands and forearms. Consistent with the results from previous reports [21,22], any history of trauma was recalled only in 45.5% of the cases. This may be related to a trauma insensitivity when engaged in agricultural activities. Except for three cases of infection via penetration of wood splinters, two cases had history of insect bite which may be a route of inoculation of soil or vegetative matter contaminated by brown pigmented fungi. Recently, CBM has been increasingly

reported in immunocompromised individuals, while diabetes is the most common comorbidity in Taiwan [18]. In our series, diabetes, and kidney health problems were the predisposing conditions.

The clinical features of CBM are the manifestation of disease evolution. Initially, the lesion is a small, nonpruriginous with erythematous papules, then the lesions progress into an erythematous plaque, with or without scales or ulcerations, and with a well-defined border. Later, the plaque expands centrifugally and develops an irregular verrucous or papillomatous surface [24]. The most frequent clinical aspect of CBM is different according to various reports. However, verrucous lesions [21,25,26], nodular lesions [25,27], cicatricial lesions [2], and plaque-like lesions [3] are respectively predominant. In our study, plaque lesions were the most common, followed by verrucous lesions. According to Queiroz-Telles et al [11], lesions can be graded as mild forms (a solitary plaque or nodule measuring less than 5 cm in diameter); moderate forms (solitary or multiple lesions which may be nodular, verrucous or plaque types, existing alone or in combination, covering one or two adjacent cutaneous regions, measuring less than 15 cm in diameter); and severe forms (any type of lesion alone or in combination, covering extensive cutaneous regions whether adjacent or non-adjacent). Severe lesions tend to respond slowly or even become non-responding to antifungal drugs. Table 1 shows that mild form is the most common in our study. Obviously, cases with severe forms had a longer disease course.

Chromoblastomycosis (CBM) is a neglected tropical disease (NTD) [12]. Its diagnostic characteristic is the presence of brown muriform cells in the infected tissue, which can be demonstrated using potassium hydroxide (KOH) mount and hematoxylin and eosin staining [28]. The sensitivity of direct examination ranges from 90 to 100% and this method is fast, easy, and inexpensive [14]. In the current study, microscopy showed muriform cells with or without germinating hyphae in the lesion scales of all patients. The tissular response is not specific in CBM specimens, and it may be similar to that of the tissue reactions observed for most implantation mycoses [14]. Pires et al. found two main types of granulomatous tissue reactions: suppurative granuloma with abundant fungal cells, mostly from verrucous lesions, and tuberculoid granuloma, with few parasites, from plaque and atrophic lesions [29]. Histopathology showed that muriform cells are present in 45.4% of the cases, while it was noted be 100% in the cases reported by Mead and Ridley in 1957 [30], and in 12 of the 13 cases that were reported by Leslie and Beardmore in 1979 [31]. CBM lesions are chronic, indolent, and clinically polymorphic. CBM can mimic a wide spectrum of diseases with infectious and noninfectious causes [14], such as cutaneous tuberculosis, leprosy, leishmaniasis, sporotrichosis, mycetoma, psoriasis, and malignancies such as verrucous carcinoma and cutaneous lymphoma [32]. Our study confirmed that 72.7% (8/11) had been misdiagnosed. This suggests the importance of improving the diagnostic ability of clinicians in rural areas, especially when using a direct microscopic examination. The fluorescent reagent was demonstrated to increase the sensitivity of the detection of many fungi, but the utility in the case of pigmented fungi is not helpful when compared with KOH.

CBM is difficult to treat and is associated with low cure rates and high relapse rates, especially in chronic and extensive cases. The treatment choice and the results depend on the etiological agent, size, extent of the lesions, topography, and the presence of complications [33]. Treatment may be divided into three groups; physical treatment, chemotherapy and combination therapy [11]. Patients showing severe and advanced clinical forms of disease require a long duration of continuous systemic antifungal treatment [34,35]. Itraconazole is the most common choice of first-line agents [18] and was the most common choice of first-line agent in our practice (200–400 mg/day). Our study showed that mild cases respond well to systemic agents, with a treatment duration varying from 3-5months. Case 3 treatment suggested that

severe lesions tend to respond slowly or even become non-responding to antifungal drugs. The 5-aminolevulinic acid-based photodynamic therapy (ALA-PDT) directly inactivate F. monophora through a ROS dependent oxidative damage and indirect activation of macrophages [36]. Case 5 treatment suggested that the combination of antifungal drugs and PDT is effective for severe forms. CBM mainly affects agricultural workers and rural population who are often at the bottom of society and unable to bear the burden of long-term treatment. In our series, it is a pity that case 1 lost the capacity to work due to the long-term lack of effective treatment. The burden and medical impact of this implantation mycosis are certainly underestimated. The use of protective equipment such as gloves, shoes, and adequate clothes may be a key in reducing the risk of infection [14].

In conclusion, CBM is a neglected fungal disease that mainly affects low-income agricultural workers and rural population. The high misdiagnosis rate of chromoblastomycosis is a major challenge in dermatology, especially for rural hospitals. The diagnosis of CBM requires laboratory confirmation by direct mycological examination and/or histopathology. The visualization of muriform cells in clinical samples is a cornerstone of this disease diagnosis. Direct microscopic examinations should be further promoted in rural hospitals. The treatment choice and results also depend on the patients' lesion grade. For severe forms, high doses and a long treatment and combination therapy are the first treatment choices. Apart from morphological identification, molecular analysis is important, especially for *Fonsecaea spp*. *F. monophora* is identified as the major pathogen affecting the hands and forearms of patients in Zhejiang. However, due to the study small case series, the results may have some limitations.

## Acknowledgments

The authors would like to express their gratitude to EditSprings (https://www.editsprings.cn) for the expert linguistic services provided.

## Author Contributions

**Conceptualization:** Xiujiao Xia.

**Data curation:** Sujun Liu, Huilin Zhi.

**Formal analysis:** Sujun Liu, Huilin Zhi.

**Funding acquisition:** Zehu Liu.

**Investigation:** Sujun Liu, Xiujiao Xia.

**Methodology:** Sujun Liu, Xiujiao Xia.

**Project administration:** Zehu Liu, Xiujiao Xia.

**Resources:** Huilin Zhi, Hong Shen, Wenwen Lv, Bo Sang, Qiuping Li, Yan Zhong, Zehu Liu.

**Supervision:** Xiujiao Xia.

**Writing – original draft:** Sujun Liu.

**Writing – review & editing:** Sujun Liu, Huilin Zhi, Hong Shen, Wenwen Lv, Bo Sang, Qiuping Li, Yan Zhong, Zehu Liu, Xiujiao Xia.

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
