## [Decision Letter · Decision Letter 0]

1 Jul 2022

Dear Mr xia,

Thank you very much for submitting your manuscript "Chromoblastomycosis: A case series from eastern China" for consideration at PLOS Neglected Tropical Diseases. As with all papers reviewed by the journal, your manuscript was reviewed by members of the editorial board and by several independent reviewers. In light of the reviews (below this email), we would like to invite the resubmission of a significantly-revised version that takes into account the reviewers' comments. 

This relatively small case series has merit as this disease has rarely been diagnosed from this part of China before, cases came forward as part of a campaign, the initial diagnosis was made with KOH, not biopsy, and most of the isolates were identified as F monophora.

The article would be stronger if these points were emphasised (along with the long period of missed diagnosis in many of them). It would also be stronger if a proper description of the campaign to find cases was described, and the results of this campaign in terms other NTDs described. how many cases of leprosy, sporotrichosis, mycetoma, cutaneous mycobacterial infections etc were found alongside chromoblastomycosis. 

The article requires extensive revision and better quality pictures (and preferably before and after treatment pictures) to be acceptable for publication. Please attend to the referees comments.

We cannot make any decision about publication until we have seen the revised manuscript and your response to the reviewers' comments. Your revised manuscript is also likely to be sent to reviewers for further evaluation.

Sincerely,

David W. Denning

Guest Editor

Ahmed Fahal

Deputy Editor

This relatively small case series has merit as this disease has rarely been diagnosed from this part of China before, cases came forward as part of a campaign, the initial diagnosis was made with KOH, not biopsy, and most of the isolates were identified as F monophora.

The article would be stronger if these points were emphasised (along with the long period of missed diagnosis in many of them). It would also be stronger if a proper description of the campaign to find cases was described, and the results of this campaign in terms other NTDs described. how many cases of leprosy, sporotrichosis, mycetoma, cutaneous mycobacterial infections etc were found alongside chromoblastomycosis. 

The article requires extensive revision and better quality pictures (and preferably before and after treatment pictures) to be acceptable for publication. Please attend to the referees comments.

Reviewer's Responses to Questions

**Key Review Criteria Required for Acceptance?**

**Methods**

-Are the objectives of the study clearly articulated with a clear testable hypothesis stated?

-Is the study design appropriate to address the stated objectives?

-Is the population clearly described and appropriate for the hypothesis being tested?

-Is the sample size sufficient to ensure adequate power to address the hypothesis being tested?

-Were correct statistical analysis used to support conclusions?

-Are there concerns about ethical or regulatory requirements being met?

Reviewer #1: Please explain in more detail what is meant by a consultation campaign.were there regional

organised clinics specifically seeking cases of chromoblastomycosis or skin disease in general. We

need to know a bit more about this recruitment method.

Was the mycological work, including direct microscopy carried out in a single laboratory ?

Reviewer #2: Suggestions and or comments were attached

Reviewer #3: The authors retrospectively reviewed 11 cases of CBM in one center of Eastern China in three years.

Reviewer #4: The objective can be more clearly stated. It is a prospective study but in the introduction, it stated that "In order to evaluate the clinical characteristics of CBM in Zhejiang,eastern China,we reviewed 11 cases..," which may imply that it was a retrospective study instead.

In page 2, paragraph 1, the author stated: "Patients were recruited between January 2018 and December 2021 and during consultation campaigns conducted in Zhejiang Province.." What was meant by consultation campaigns? Were these "information campaigns" conducted in other clinics so that these kinds of cases were referred to the Hangzhou Third people's Hospital? It may good to describe further the "consultation campaign" and referral system followed since this would give readers an idea on the actual catchment area of the facility. Is Hangzhou Third people's Hospital a specialty hospital?

Since it is a case series, conducted prospectively, it would have been better if pictures of individual cases were taken, and sizes of lesions were recorded. It would be interesting to note if cases with smaller lesion sizes responded to therapy faster and more completely, compared to those with larger and more disseminated lesions. 

Was surgery an option in any of the cases? If not, why not?

**Results**

-Does the analysis presented match the analysis plan?

-Are the results clearly and completely presented?

-Are the figures (Tables, Images) of sufficient quality for clarity?

Reviewer #1: Please explain the following phrase - consolidate treatment due to the interference of comorbidity.

Kidney trouble – better chronic renal disease.

Reviewer #2: The use of lesions severity graduation would be this article more interesting for clinicians

Reviewer #3: The study indicated that the most common pathogenic fungus was F.monophora.

Reviewer #4: Figure 1 clinical photos are not clear. It would be good to indicate which particular cases these were.

Since it is a case series, it would be good to include before and after treatment pictures as well.

**Conclusions**

-Are the conclusions supported by the data presented?

-Are the limitations of analysis clearly described?

-Do the authors discuss how these data can be helpful to advance our understanding of the topic under study?

-Is public health relevance addressed?

Reviewer #1: The authors argue for improving diagnostic ability in rural areas to reduce the risk of misdiagnosis. It

would, be useful here to discuss possible methods of doing this.

Please mention other forms of treatment such as oral terbinafine or heat therapy.

It would be useful in the discussion to on the distribution and case load of chromoblastomycosis in

China. How common is it ? Are there any explanations for the differ ent distribution of organisms ?

Why is F monophora a dominant cause in China ? Is this a true geographic variation or is this because

without molecular tools most such strains have been misidentified as F pedrosoi in the past ?

Reviewer #2: Suggestions and or comments were attached

Reviewer #3: The authors should indicate the limitations of this case series, such as the limited number of the patients, single center study, etc.

Reviewer #4: The conclusions can be improved to refer more to the cases seen in the facility.

**Editorial and Data Presentation Modifications?**

Reviewer #1: There are some changes needed to the English. Please mention that chromoblastomycosis is

formally designated by WHO as an NTD

Reviewer #2: (No Response)

Reviewer #3: None

Reviewer #4: Page 4, paragraph 1: associated condition, not predisposing

Page 3, paragraph 1: 8 cases were misdiagnosed; Page 4, paragraph 2: 7 cases were diagnosed. Please clarify.

**Summary and General Comments**

Reviewer #1: (No Response)

Reviewer #2: Suggestions and or comments were attached

Reviewer #3: The novelty of this study is relatively low.

Reviewer #4: 1. There may be a need to edit the article for it to be publishable.

2. These are the points that I think are good take-aways from this article

a. All cases were initially evaluated in other clinics before the information campaign and 8 were misdiagnosed. This highlights the importance of educating primary physicians or those who usually initially manage these cases, on recognizing chromoblastomycosis and other skin neglected tropical diseases.

b. All 11 cases were diagnosed with chromoblastomycosis after doing a simple, easy, non-invasive, inexpensive, out-patient procedure, the KOH smear. This illustrates that KOH smear may even be more sensitive in catching these cases compared to punch biopsy.

PLOS authors have the option to publish the peer review history of their article (what does this mean?). If published, this will include your full peer review and any attached files.

Reviewer #1: No

Reviewer #2: Yes: Flávio Queiroz-Telles

Reviewer #3: No

Reviewer #4: Yes: Maria Christina Filomena Batac
---

## [Decision Letter · Decision Letter 1]

16 Aug 2022

Dear Mr xia,

Thank you very much for submitting your manuscript "Chromoblastomycosis: A case series from eastern China" for consideration at PLOS Neglected Tropical Diseases. As with all papers reviewed by the journal, your manuscript was reviewed by members of the editorial board and by several independent reviewers. The reviewers appreciated the attention to an important topic. Based on the reviews, we are likely to accept this manuscript for publication, providing that you modify the manuscript according to the review recommendations. 

Please remove the term dematiaceous and replace with brown pigmented - throughout the manuscript, including the abstract.

There are still some errors of english, which should be improved before final publication. For example 'histopathology, not histopathology of biopsies'

In the abstract just state how many males, not a ratio.

The authors have added helpful information on the other similar presentations. Do they have any sense of trends over time? Is CBM diminishing in frequency, or static? And how does it compare to sporotrichosis, for example.

Sincerely,

Ahmed Fahal, FRCS, FRCSI, FRCSG, MS, MD, FRCP(London)

Section Editor

Ahmed Fahal

Section Editor

Please remove the term dematiaceous and replace with brown pigmented - throughout the manuscript, including the abstract.

There are still some errors of english, which should be improved before final publication. For example 'histopathology, not histopathology of biopsies'

In the abstract just state how many males, not a ratio.

The authors have added helpful information on the other similar presentations. Do they have any sense of trends over time? Is CBM diminishing in frequency, or static? And how does it compare to sporotrichosis, for example.

Reviewer's Responses to Questions

**Key Review Criteria Required for Acceptance?**

**Methods**

-Are the objectives of the study clearly articulated with a clear testable hypothesis stated?

-Is the study design appropriate to address the stated objectives?

-Is the population clearly described and appropriate for the hypothesis being tested?

-Is the sample size sufficient to ensure adequate power to address the hypothesis being tested?

-Were correct statistical analysis used to support conclusions?

-Are there concerns about ethical or regulatory requirements being met?

Reviewer #1: The authors have answered my queries

Reviewer #2: The author added the reviwers suggestions or comments to the current manuscript version

Reviewer #4: The paper described that aside from chromoblastomycoses, other subcutaneous mycoses were diagnosed during the same 3-year period of the study. It is good to disclose if these have been described or will be described in another paper, (and include proper citations).

**Results**

-Does the analysis presented match the analysis plan?

-Are the results clearly and completely presented?

-Are the figures (Tables, Images) of sufficient quality for clarity?

Reviewer #1: The authors have answered my queries

Reviewer #2: The author added the reviwers suggestions or comments to the current manuscript version

Reviewer #4: Pictures had no accompanying descriptive texts.

It will be good to compare the frequency of cases of subcutaneous mycoses diagnosed over the years in their facility. Has there been an increase in diagnosis due to greater awareness of the disease entities? Or after they have adapted this "routine flowchart" in diagnosing subcutaneous diseases?

**Conclusions**

-Are the conclusions supported by the data presented?

-Are the limitations of analysis clearly described?

-Do the authors discuss how these data can be helpful to advance our understanding of the topic under study?

-Is public health relevance addressed?

Reviewer #1: The authors have answered my queries

Reviewer #2: This paper is relevant for the Public Health because chromoblastomycosis is one of the few fungal infections, officially recognized as NTD by the WHO

Reviewer #4: (No Response)

**Editorial and Data Presentation Modifications?**

Reviewer #1: Nil needed

Reviewer #2: Please note that "dematiaceous fungi "is an obsolete term. "Melanized fungi" is week stablished denomanation for all chromoblastomicosis and phaeohyphomiycosis etiological agents

Reviewer #4: Abstract: Since the sample being described is small, only 11, then it is better to just say outright how many are males and how many are females, instead of giving the ratio of 4.5:1

Terms: histopathology, not histopathology of biopsies

**Summary and General Comments**

Reviewer #1: The authors have answered my queries

Reviewer #2: I suggested a mino revision

Reviewer #4: The flow of the narrative can be improved. Although much improved from the first draft, it needs more editing to be publishable.

PLOS authors have the option to publish the peer review history of their article (what does this mean?). If published, this will include your full peer review and any attached files.

Reviewer #1: No

Reviewer #2: Yes: Flavio Queiroz-Telles

Reviewer #4: Yes: Maria Christina Filomena Batac

Figure Files:

Data Requirements:

Reproducibility:

References

---

## [Editor Report · Decision Letter 2]

7 Sep 2022

Dear Mr xia,

We are pleased to inform you that your manuscript 'Chromoblastomycosis: A case series from eastern China' has been provisionally accepted for publication in PLOS Neglected Tropical Diseases.

Best regards,

David W. Denning

Guest Editor

Ahmed Fahal

Section Editor

None

---

## [Editor Report · Acceptance letter]

14 Sep 2022

Dear Mr Xia,

We are delighted to inform you that your manuscript, "Chromoblastomycosis: A case series from eastern China," has been formally accepted for publication in PLOS Neglected Tropical Diseases.

Best regards,

Shaden Kamhawi

co-Editor-in-Chief

Paul Brindley

co-Editor-in-Chief
